# What Color Does the Consumer See? Perceived Color Differences in Plastic Products in an LED-Lit Environment

**Xiao Dou** [1]**, Chih-Fu Wu** [2,*]**, Kai-Chieh Lin** [2] **and Jeih-Jang Liou** [3]

1    The Graduate Institute of Design Science, Tatung University, Taipei 104, Taiwan; michelledou007@outlook.com
2    Department of Industrial Design, Tatung University, Taipei 104, Taiwan; kclin@ttu.edu.tw
3    Department of Product and Media Design, Fo Guang University, Yilan 262, Taiwan; jjliou@mail.fgu.edu.tw
*    Correspondence: wcf@gm.ttu.edu.tw

**Abstract:** To attract customers and increase market opportunities, retailers frequently use lighting to highlight the color of their products. However, differences between perceived and actual color, triggered by display lighting, can motivate buyers to discard products after purchase. Few studies have been reported on differences in perceived color, caused by LEDs. This study focuses on two correlated color temperatures (2800 K, 4000 K) and illuminance levels (500 lx, 1500 lx) to create four LED-lit environments, and measures the differences in the color perceived by 20 observers on acrylonitrile–butadiene–styrene (ABS) plastics, with different surfaces, under these four environments. The results reveal that correlated color temperature results in larger perceived differences in color than illuminance, and the effects of LED light sources on green and yellow ABS plastic products are more obvious than their effects on red and blue products. One possible reason for this can be attributed to the visual sensitivity effect of human eyes. The results of this study can serve as a reference for designers fabricating ABS plastic products for practical lighting applications, and improving the role of LED lighting in sustainable development.

**Keywords:**    Light-emitting diode (LED); Perceived color difference; Lighting design; Acrylonitrile–butadiene–styrene (ABS) plastic

## 1. Introduction

The color of products and the lighting atmosphere are crucial aspects of design, with the aim of attracting customers and increasing market opportunities [1,2]. During the first 90 seconds of consumers' initial interactions with a product, approximately 62%–90% of their assessment is based on its colors alone [3]. Furthermore, 85% of consumers claimed that color was the reason they decided to buy a particular product [3]. Color-based design strategies typically produce the lowest-cost sales strategies for such products and, as such, are highly cost-effective [4]. Lighting atmosphere in stores is used to convey a certain shopping experience and increase sales [2,5]. To create an optimal atmosphere, retailers frequently use lighting to highlight the color of products and encourage customers to purchase them [6]. However, if display illumination is misused, it might cause unsustainable consumption behaviors [7,8]. Customers buy products they do not need more freely when influenced by certain lighting atmospheres in a store, a process known as over-consumption [8,9]. Low-correlated color temperatures in lighting lead to less rational decision-making [10,11], excitement, and a longer amount of time spent in retail outlets [12]. This is particularly apparent in the purchase of cheap, low-durability products (e.g., plastic storage boxes). An even more critical factor is that, because the colors that consumers see in store are the mutually influenced result of the colors of products and lighting, as

soon as consumers actually use the products, they may realize that the products' colors are different from the color they perceived in the store. This discrepancy leads to a low attractiveness level of the purchased product and an increased risk of the consumer discarding it [13,14]. To solve this problem, a balance should be found for display illumination in stores between creating a target atmosphere and reducing lighting-triggered color differences in products.

Light-emitting diodes (LEDs) are the preferred means of display illumination in shopping malls because of their high energy efficiency, low maintenance cost, longevity, low pollution, and ease of adjustment [15–17]. However, because of the unique characteristics of LED lights, their use inside stores may result in greater differences in color [18]. LED light sources have three functions in product displays: color rendering, correlated color temperature, and illumination [19]. An LED's color rendering is set at the factory, and, the higher the index, the better the display effect. Thus, color rendering has rarely been manipulated in experiments [20], and previous studies mainly focused on controlling the correlated color temperature (CCT) and illuminance. Such studies determined that CCT and LED illuminance affects visual perception. Light source illuminance greatly affects the human eye's discrimination of spatial objects [21], and a light source's correlation color temperature can significantly affect a person's color discrimination ability [22]. Hawes, Brunyé [23] reported that an increase in color temperature enhances positive emotions and wakefulness, thus expediting the cognitive task of visual perception. In practice, the choice of lighting system has considerable effects on the space that it illuminates; this, in turn, affects customer psychology, in relation to the colors of products in that space. Thus, appropriate illuminance is key when minimizing color differences [24]. Relative to their counterparts, LED light sources are easy to adjust. Therefore, if color difference tendencies can be predicted, lighting source-induced color difference can be reduced. This not only saves energy with respect to the light source but also reduces an unpleasant experience for shoppers, caused by the discrepancy between a product's color when displayed and its color when used.

Plastic products and packaging are common sights in people's daily lives, and have become a symbol of "unsustainable" consumption in industrialized countries [25,26]. Plastic is commonly used in product design, ranging from automobiles and electronic devices to small plastic containers and packaging [26]. Acrylonitrile–butadiene–styrene (ABS) polymers are preferred plastic materials, due to their high impact resistance, chemical stability, satisfactory electrical performance, flame retardancy, transparency, high flexibility, and high heat resistance [27,28]. Typically, plastics (ABS) used in product design undergo a spray-paint surface treatment, and different surfaces (e.g., surface gloss) can influence perception of their color [29,30]. The appropriate bright spray paint and matte spray paint are applied, depending on the gloss used [31]. Gloss (the ability to reflect reflected light) is an important characteristic of spray-painting. High gloss is called bright spray paint, and the inverse is matte spray paint. Bright spray paint is typically used in automobile design [32], and matte spray paint normally used in household application design. Accurate color replication is crucial once a design is completed, thus, color systems have been designed. This study used the Natural Color System (NCS), a quantitative model that corresponds to human perception of color [22,33]. The NCS has 1950 colors, and each color encodes three characteristics: blackness, saturation, and hue [33]. For example, the color S2060-G10Y represents a color with 20% blackness, 60% saturation, and a hue of 10% yellow and 90% green.

Huang, Liu [34] explored consumers' preferences for jeans exposed to different CCTs. Wu, Wu [35] studied the color discrimination of coated paper and polyester fiber under different light sources. Tantanatewin and Inkarojrit [1] reported that lighting conditions significantly affect perception of retail identity. Shih [10] discussed the influence of light sources on consumers' perceptions of a fruit's color. Their results indicated that, although different light source designs produce different display atmospheres, any atmosphere is liable to produce color differences. However, these studies did not consider the influence of light source parameters differences in human perception of color.

Therefore, this study focused on the influence of the interaction between LED light source parameters and product color on differences in visual perception of color. Specifically, color differences,

in relation to ABS matte and bright spray-painted products under commonly used illuminations and CCTs, were investigated. The trends in perceived differences in color were also predicted. The results of this study serve as adjustment references for the actual design of LED light sources, allowing the intended display effect to be realized while simultaneously reducing color differences, thereby optimizing the role of LED lighting in sustainable development.

## 2. Materials and Methods

### 2.1. Experimental Design

#### 2.1.1. Experimental Light Source Setting

In this study, the changes in perceived color of different colors of bright and matte spray-painted ABS plastic products, induced by different LED-lit environments, were discussed. The experiment primarily controlled for CCT and illuminance. Sayigh [36] reported that the CCTs of a light source in a general store display are typically in the 2800–4000 K range. Lighting at 2800 K CCT is considered warm, and that at 4000 K CCT is considered white [37]. In this study, 2800 K and 4000 K CCTs were selected. According to Chinese National Standards (CNS) [38] and the Japanese Industrial Standards (JIS) table of illumination [39], the illumination level of large stores (such as department stores) is typically 500 lx, and that in the windows and design counters of major large stores is typically 1500 lx. The two illumination levels used in this study's experiment were 1500 lx and 500 lx.

To meet the levels of experimental variables, a customized experimental light box of two color temperatures * two illumination levels was used in the experiment, and produced four LED-lit environments. An LED lamp comprises five LED modules, each of which involves a combination of multiple LED lights. Stores with higher color requirements will choose a lighting color rendering index level of 1B (Ra: 80–90) [40]. In this study, LEDs with Ra = 85 were selected. An illuminance meter (TenmarsTM202) was used to correct the light source, and its illuminance range was located within 20 lx.

#### 2.1.2. Stimuli

The four selected colors were green, yellow, blue, and red [41]. In the NCS color system, black and white constitute the central axis, which is surrounded by a hue ring comprising green (G), yellow (Y), blue (B), and red (R). Thus, green, yellow, blue, and red were selected as experimental colors in this experiment. According to the NCS color system, the color stimuli numbers of the four colors were as follows: Green = S2060-G10Y, Yellow = S0575-G90Y, Blue = S1565-B, and Red = S1080-Y90R. For each of the four standard color stimuli, 13 comparison color stimuli were selected, in the three directions of blackness, saturation, and hue (Figures 1–4). Previous research has demonstrated that the saturation of green is positively correlated with the four light conditions used in this paper [42]. Therefore, the comparison green stimulus had a higher saturation value than the standard stimuli. The stimuli to be tested in the experiment were painted on white ABS plastic for customization. The gloss of matte spray paint was 13, and the gloss of bright spray paint was 88. The stimuli had a size of 2*2 cm$^2$, which corresponds to 2° of the viewing field at a given viewing distance, as reported in the experiment of Pardo, Cordero [22]. The spectral reflectance distribution (SRD) of eight standard stimuli (four colors × two spray painting types) in the visible spectrum were measured using a calibrated spectrophotometer (X-Rite i1) [34], and shown in Figure 5.

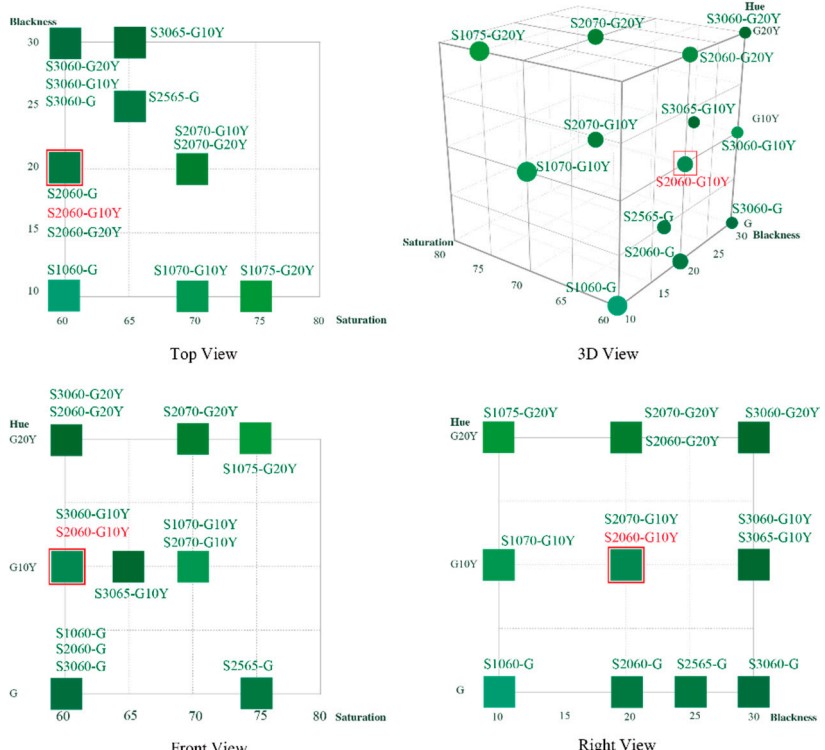

**Figure 1.** Standard and comparison stimuli (green).

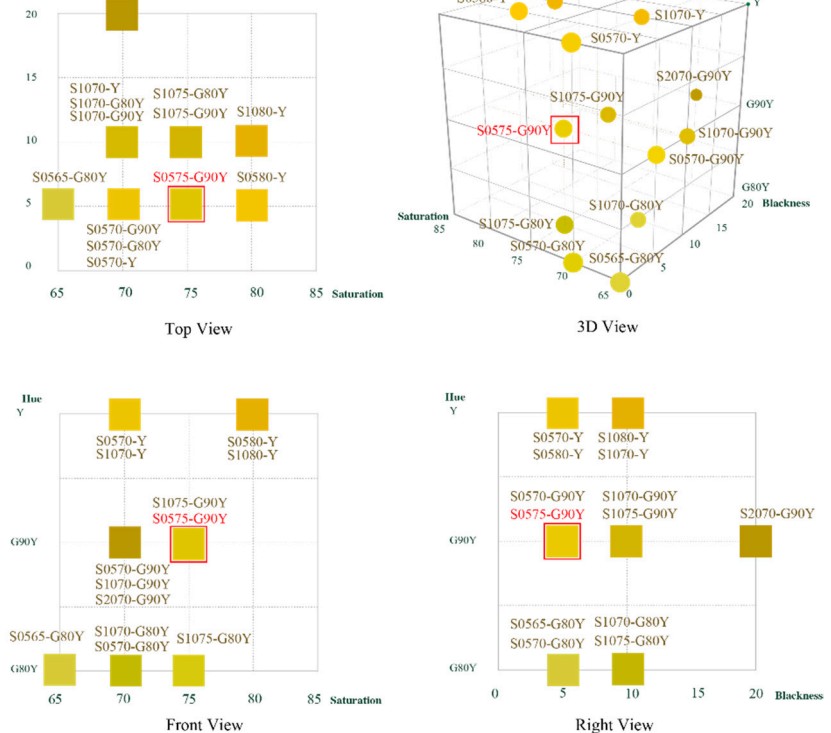

**Figure 2.** Standard and comparison stimuli (yellow).

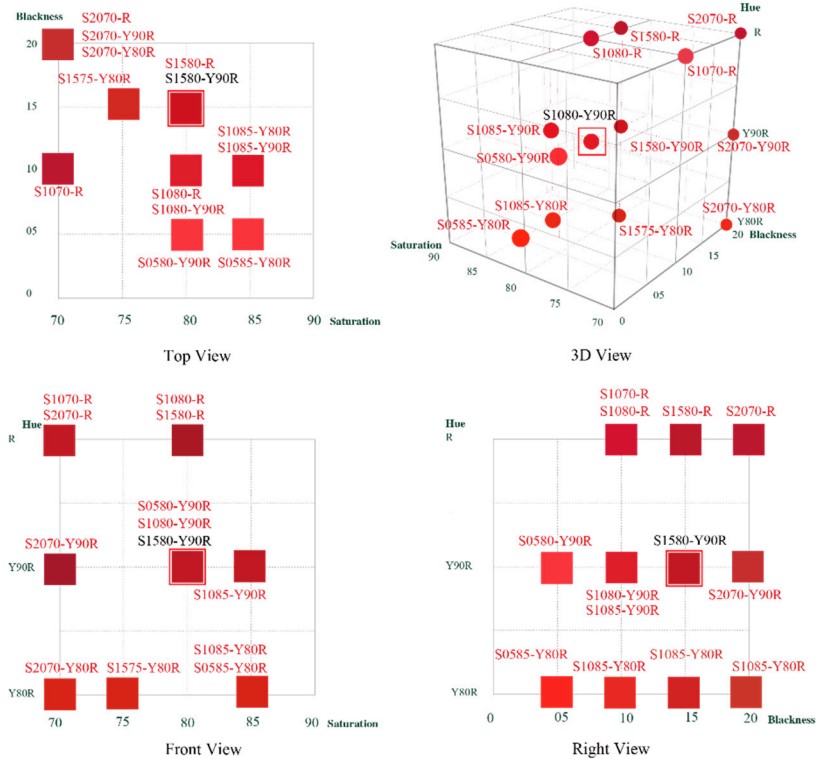

**Figure 3.** Standard and comparison stimuli (red).

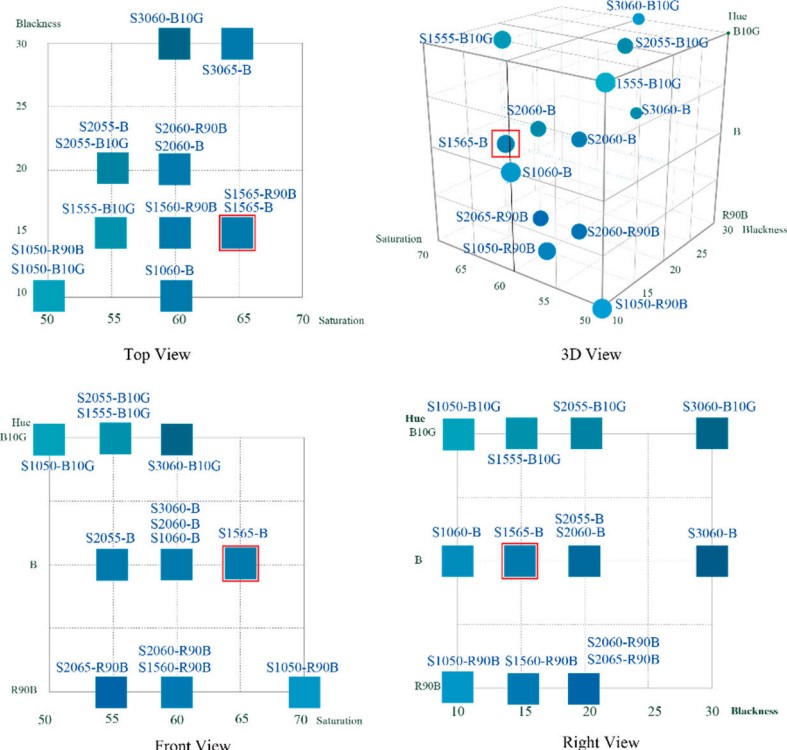

**Figure 4.** Standard and comparison stimuli (blue).

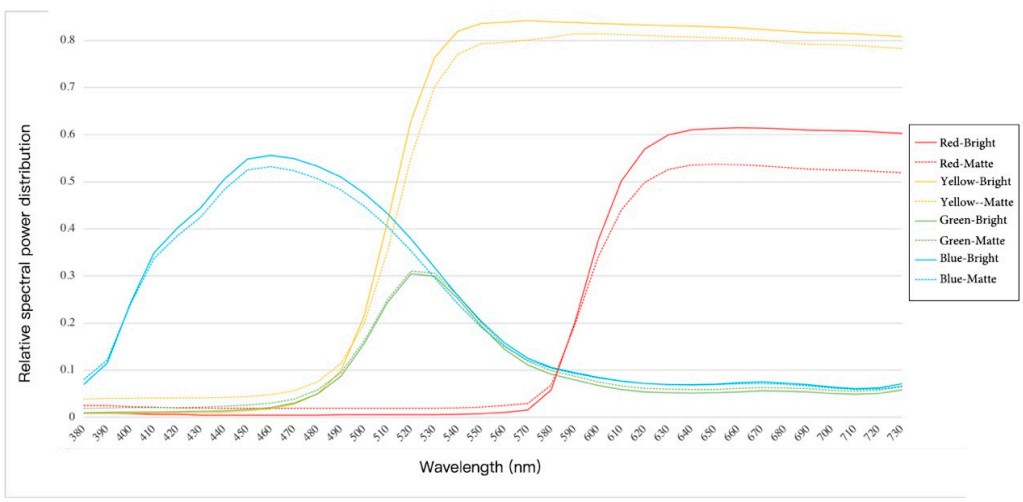

**Figure 5.** Spectral reflectance distribution.

### 2.1.3. Variables

In the experiment, a 2 × 2 × 2 × 4 full-factor, within experimental design, was adopted, and the experimental variables and levels are presented in Table 1.

**Table 1.** Experimental variables.

| Variables | Level |
|---|---|
| Correlated color temperature (CCT) | 2800 K<br>4000 K |
| Illuminance level | 500 lx<br>1500 lx |
| Spray paint types | Matte spray paint, gloss = 18<br>Bright spray paint gloss = 88 |
| Product color | Green (G) = S2060-G10Y<br>Yellow (Y) = S0575-G90Y<br>Red (R) = S1080-Y90<br>RBlue (B) = S1565-B |

### 2.2. Experimental Setups

Figure 1 depicts the relative positions of the apparatus. To reduce ghosting effects due to ambient light, which affect the measurements, light pollution was eliminated in the experimental room. Moreover, to reduce the influence of temperature and humidity, we set the temperature of the space to 26 °C (which is optimal for comfort) and the humidity to 60%. We also ensured that the air conditioner was not blowing directly into the participants' faces, thus mitigating dry eye. To determine the most effective distance from the target, we measured the distance between the table and the eyes of two participants in a pretest. Because intense glare results if LED light is shone directly into a participant's eyes, the visual distance was set at 42.3 cm. The chair was adjustable in height, and each participant adjusted the chair height to ensure that the visual angle was within 30° in accordance with values in the literature [43].

The sizes of the experimental LED light box and the standard light box (the light source was D65, which is international standard artificial daylight) that emitted the comparison color stimuli were identical (Figure 6). LED illuminance and the corresponding color temperature had to be adjusted during the experiment. The spectral power distribution (SPD) of the experimental LED light source is illustrated in Figure 7.

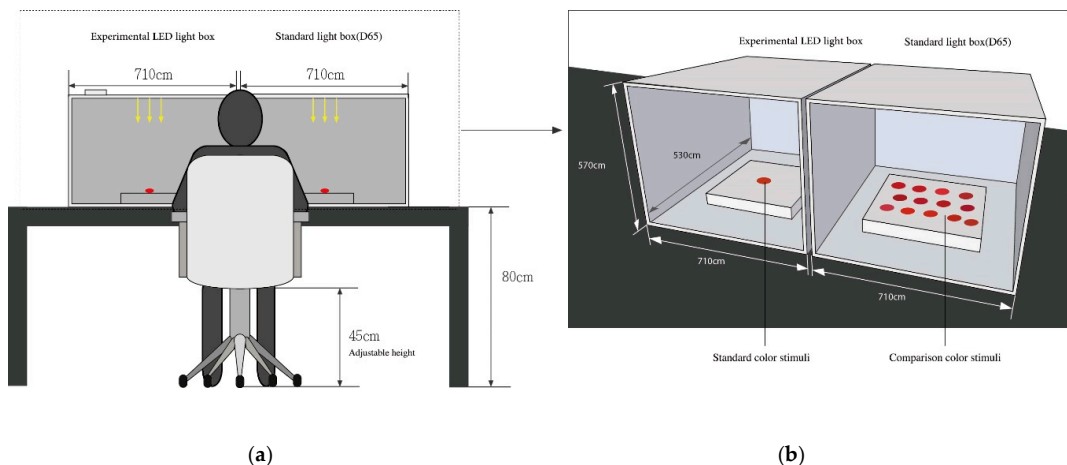

|             |             |
| :---------: | :---------: |
| (**a**)     | (**b**)     |

**Figure 6.** Experimental setups: (**a**) Front view, (**b**) Enlarged view.

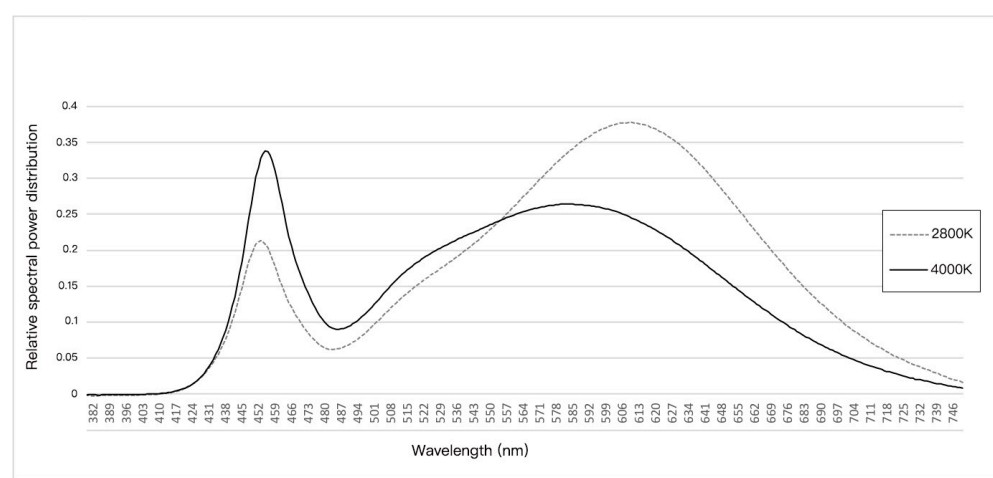

**Figure 7.** Spectral power distribution of the experimental light source.

### 2.3. Participants

In total, 26 people (12 men and 14 women) participated in the pretest, and 20 people (eight men and 12 women) participated in the experiment. The participants' visual acuities after correction were determined to be greater than 0.8, according to the Snellen's E chart. The mean age was 27, with a standard deviation of 3.45. Participants were determined to have no color blindness or color weakness, using the Ishihara Test for Color Blindness [44].

### 2.4. Measurement

The CIEL*C*h values of the selected comparison stimulus were measured. CIEL*C*h was used because it represents the color perceived by the human eye. In Commission Internationale de l'Eclairage (CIE), color space, L* (lightness), represents brightness (higher values indicate higher brightness), C* (chroma), represents chroma (higher values indicate greater chroma), and h represents hue angle. Each NCS standard and comparison color stimulus was written on the back of the stimulus, according to the color's number, blackness, saturation, hue, and corresponding CIEL*C*h space coordinate values, which were converted under D65, for 2° of the viewing field (Figure 8).

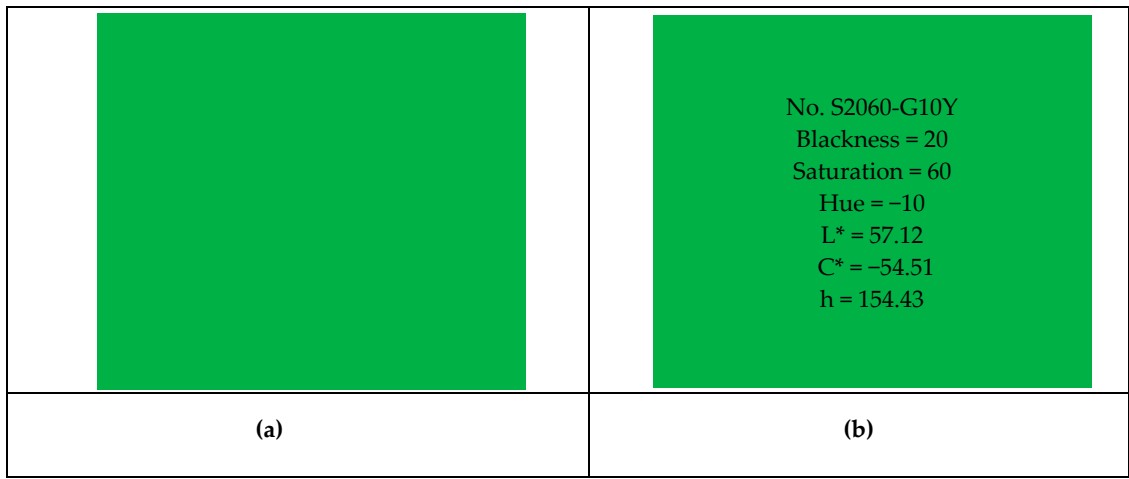

**Figure 8.** (**a**) Front, (**b**) Back.

## 2.5. Procedure

In order to eliminate the problem of color discrimination in the test subjects, the participants underwent an visual acuity and color blindness examination before entering the formal experiment. Each participant was instructed to sit on a chair and take a short rest before the experiment, to relax their eyes. Next, the participant's view angle was adjusted, to ensure that the participant was in the appropriate posture and range for color judgment. The participant was then instructed to complete a questionnaire, providing basic personal information and reading an explanation of the experimental procedure, as well as a definition of related terms. Once the questionnaire was completed, the objective of the experiment was explained again, to confirm that the participant understood the experimental procedure, after which the experiment was initiated.

1. The experiment was conducted over a course of four days, and each participant was exposed to only one color (i.e., green, yellow, blue, or red) each day and in random order between participants. Each experiment was conducted as follows:
2. The light source was turned on and left to stabilize. A photometer and colorimeter were used, to ensure that the settings were within the designated values, before starting the experiment.
3. The researcher explained the experiment procedure to the participants and provided additional information, based on the questions posed by the participants. Subsequently, the participants were asked to tie up their hair, to ensure that their eyes were completely revealed.
4. Standard color stimulus and comparison color stimulus were placed in the experimental LED light box and D65 light box, respectively. The comparison color stimulus was placed in random positions.
5. The participant selected the comparison color stimulus they believed to be closest to the standard color. The researcher then recorded the NCS color code of the selected color.
6. To prevent the learning effect, the participants was required to complete the experiment within a time limit of 2 minutes. This time limit was set on the basis of the maximum time taken by the participants during the pretest.

## 3. Results

*Multivariate Analysis of the Influence of Color, Temperature and Illumination on Color Difference*

Multivariate analysis of variance (MANOVA) was used to compare the color difference in ABS plastic paint products, as perceived by 20 participants, under different CCTs and illuminance levels.

For matte spray-painted ABS plastic products, MANOVA results (Table 2) indicated that CCT had significant effects on the perception of the lightness [F (1,80) = 25.16, p < 0.001] and chroma [F (1,80) =

9.21, p < 0.05] of green matte spray-painted plastic products. Under 2800 K CCT, the lightness and chroma of green products decreased. Illumination levels significantly affected the chroma [F (1,80) = 5.70, p < 0.05] of yellow matte spray-painted plastic products; under 1500 lx illumination, chroma increased. CCT had a significant effect on the hue angle [F (1,80) = 82.24, p < 0.05] of yellow matte spray-painted plastic products. Under 2800 K CCT, the hue angle deflected toward red. CCT had a significant effect on the lightness [F (1,80) = 29.95, p < 0.05] and hue angle [F (1,80) = 9.37, p < 0.05] of blue matte spray-painted plastic products; under 2800 K CCT, lightness decreased and the hue angle shifted to red. CCT had a significant effect on the chroma [F (1,80) = 3.92, p < 0.05] of red matte spray-painted plastic products; under 2800 K CCT, chroma increased (Figure 9).

CCT and illumination levels had a significant interaction effect on the lightness [F (1,80) = 7.81, p < 0.05] of green matte spray-painted plastic products. The lightness of 4000 K * 500 lx was higher than that of 4000 K * 1500 lx. Under 2800 K CCT, the lightness of 1500 lx was higher than that of 500 lx.

For bright spray-painted ABS plastic products, MANOVA results (Table 3) indicated that CCT had significant effects on the lightness [F (1,80) = 8.91, p < 0.001], chroma [F(1,80) = 4.52, p < 0.001], and hue angle [F(1,80) = 6.23, p < 0.001] of green bright spray-painted plastic products. Under 4000K CCT, lightness and chroma increased, and the hue angle was less shifted toward yellow. CCT had a significant effect on the lightness [F(1,80) = 6.23, p < 0.001] and hue angle [F (1,80) = 4.33, p < 0.001] of yellow bright spray-painted plastic products; under 4000 K CCT, lightness increased, and the hue angle shifted less toward red. Illumination level had a significant effect on the chroma [F (1,80) = 5.33, p < 0.05] of yellow bright spray-painted plastic products; under 1500 lx illumination, chroma increased. CCT had a significant effect on the lightness [F (1,80) = 11.25, p < 0.05] of blue bright spray-painted plastic products; under 4000 K CCT, lightness increased (Figure 10).

**Table 2.** MANOVA results for matte spray-painted products.

| | | | Means and SD | | | | F Value | | |
| | | | 4000 K | | 2800 K | | Main Effect | | Interaction Effect |
| | | D65 | 1500 lx | 500 lx | 1500 lx | 500 lx | Illuminance Level | CCT | CCT* Illuminance Level |
|---|---|---|---|---|---|---|---|---|---|
| G | L* | 57.12 | 58.17 (4.44) | 58.84 (4.19) | 56.56 (3.28) | 53.20 (3.88) | 3.45 | 25.16* | 7.81* |
| | C | 54.51 | 59.07 (5.27) | 60.14 (6.29) | 57.08 (5.41) | 56.20 (4.37) | 0.01 | 9.21* | 1.00 |
| | H | 154.43 | 147.41 (5.20) | 148.55 (6.24) | 148.12 (7.54) | 146.49 (5.28) | 0.05 | 0.36 | 1.54 |
| Y | L* | 85.31 | 84.74 (3.36) | 83.34 (3.79) | 84.04 (3.17) | 81.77 (3.41) | 8.57 | 3.30 | 0.49 |
| | C | 85.72 | 82.98 (2.52) | 82.17 (2.90) | 84.02 (2.60) | 82.43 (2.99) | 5.70* | 1.65 | 0.60 |
| | H | 95.23 | 93.40 (2.05) | 93.37 (1.88) | 90.39 (2.45) | 89.59 (1.79) | 1.23 | 82.24* | 1.05 |
| B | L* | 53.66 | 54.34 (5.22) | 52.42 (5.18) | 49.67 (3.79) | 47.71 (4.53) | 5.10 | 29.95* | 0.00 |
| | C | 38.42 | 38.67 (3.63) | 37.98 (4.17) | 38.96 (4.66) | 37.95 (6.29) | 0.94 | 0.02 | 0.04 |
| | H | 252.94 | 250.71 (12.06) | 252.00 (9.69) | 257.34 (7.65) | 256.53 (10.20) | 0.02 | 9.37* | 0.33 |
| R | L* | 40.61 | 47.41 (3.56) | 47.65 (3.72) | 48.84 (2.85) | 47.23 (3.97) | 1.12 | 0.61 | 2.08 |
| | C | 73.31 | 80.20 (6.62) | 80.54 (5.28) | 84.25 (6.08) | 80.80 (5.89) | 2.04 | 3.92* | 3.04 |
| | H | 33.47 | 35.92 (3.63) | 35.31 (4.23) | 37.59 (4.07) | 35.47 (4.25) | 3.43 | 1.54 | 1.05 |

Note: Green = G; Yellow = Y; Blue = B; Red = R. * p < 0.05.

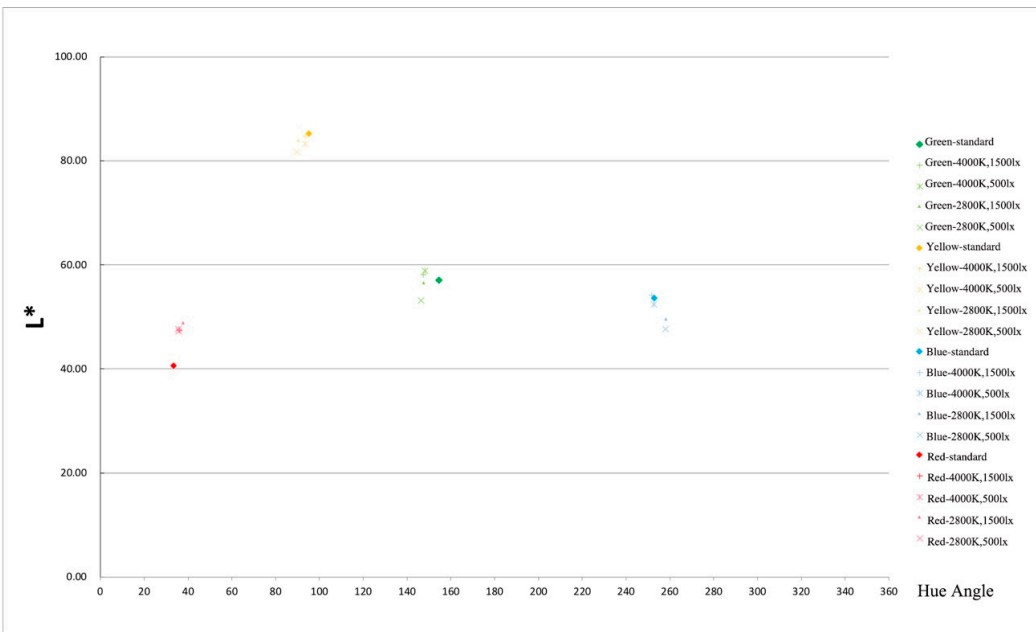

**Figure 9.** Perception of color coordinate positions of matte spray painted plastic products (green, yellow, blue and red) under four LED-lit environments.

**Table 3.** MANOVA results for bright spray-painted plastic products.

| | | | Means and SD | | | | F Value | | |
| | | | 4000 K | | 2800 K | | Main Effect | | Interaction Effect |
| | | D65 | 1500 lx | 500 lx | 1500 lx | 500 lx | Illuminance Level | CCT | CCT* Illuminance Level |
|---|---|---|---|---|---|---|---|---|---|
| G | L* | 57.12 | 55.64 (4.81) | 55.45 (6.08) | 52.91 (4.65) | 52.61 (4.84) | 0.07 | 8.91* | 0.00 |
| | C | 54.51 | 57.00 (4.74) | 56.83 (5.06) | 56.04 (5.27) | 54.00 (4.55) | 1.54 | 4.52* | 1.10 |
| | H | 154.43 | 149.33 (5.30) | 150.67 (5.19) | 148.07 (6.00) | 146.80 (7.31) | 0.00 | 5.51* | 1.43 |
| Y | L* | 85.31 | 84.26 (3.54) | 83.72 (3.62) | 83.15 (3.47) | 81.54 (3.84) | 2.64 | 6.23* | 0.66 |
| | C | 85.72 | 83.97 (2.56) | 82.81 (2.58) | 83.34 (2.66) | 82.25 (2.90) | 5.33* | 1.51 | 0.01 |
| | H | 95.23 | 92.08 (3.19) | 92.18 (2.58) | 91.94 (4.79) | 89.73 (2.64) | 2.89 | 4.33* | 3.45 |
| B | L* | 53.66 | 51.26 (6.20) | 49.48 (4.76) | 47.05 (3.16) | 47.60 (5.36) | 0.46 | 11.25* | 1.65 |
| | C | 38.42 | 40.02 (4.65) | 38.80 (4.29) | 39.58 (5.24) | 38.47 (5.02) | 1.77 | 0.19 | 0.00 |
| | H | 252.94 | 256.60 12.47 | 257.11 9.02 | 260.94 8.37 | 257.79 11.49 | 0.48 | 1.74 | 0.92 |
| R | L* | 40.61 | 45.47 (4.48) | 47.33 (3.80) | 45.15 (4.22) | 46.17 (4.24) | 3.59 | 0.94 | 0.30 |
| | C | 73.31 | 75.15 (6.50) | 77.06 (5.99) | 77.86 (5.14) | 74.29 (6.20) | 0.59 | 0.00 | 6.37* |
| | H | 33.47 | 33.19 (3.63) | 33.91 (4.23) | 35.32 (4.07) | 33.12 (4.25) | 0.98 | 0.80 | 3.77 |

Note: Green = G; Yellow = Y; Blue = B; Red = R * p < 0.05.

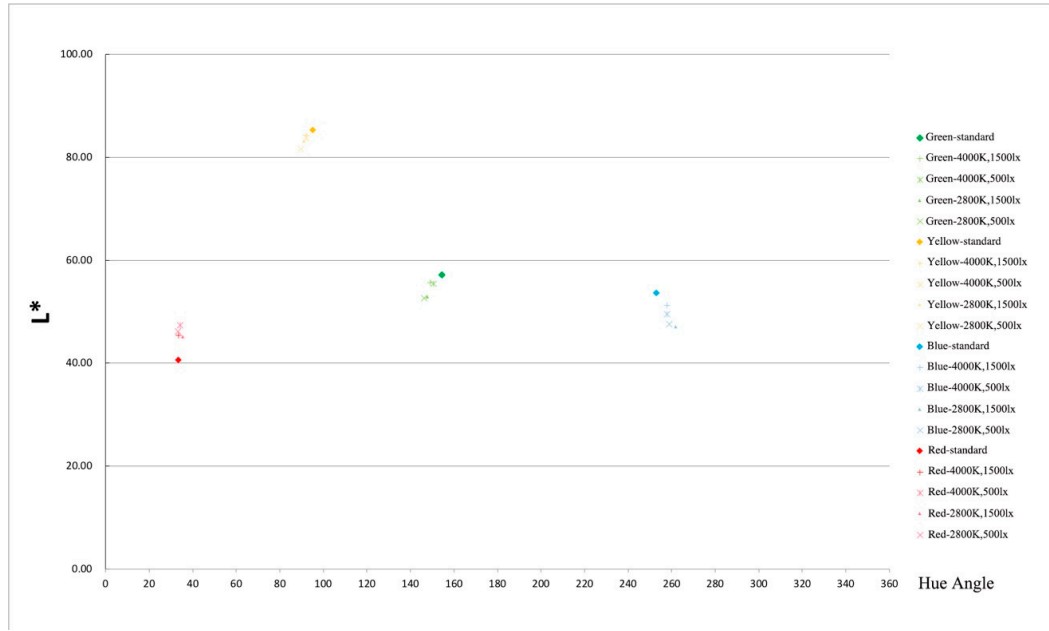

**Figure 10.** Perception of color coordinate positions of bright spray painted plastic products (green, yellow, blue, and red) under four LED-lit environments.

CCT and illumination levels had a significant interaction effect on the chroma [$F_{(1, 80)} = 6.37$, $p < 0.05$] of red products. The chroma of 4000 K * 500 lx was higher than that of 4000 K * 1500 lx. Under 2800 K CCT, the chroma of 1500 lx was higher than under 500 lx.

## 4. Discussion

The results demonstrate that the LED light source affects the color of the plastic material products. For matte spray-painted plastic products, in any of the four colors, as long as the LED illuminance level is within the legal range of 1500–500 lx, only yellow colors are affected by the illumination level. Specifically, the chroma of yellow increases with illumination. The results of this study confirm that the illumination level of LED light sources in Taiwan has a small effect on the perceived color of green, blue, and red products. However, the effects of CCT are greater than those of illumination level. If the CCT of an LED light source is within 2800–4000 K, CCT mainly affects the chroma of green and red colors, as well as the lightness of green and blue colors, consistent with the results of Wu et al., [45]. Their study determined that the lightness and chroma of green are very easily affected by the setting of the light source. Moreover, it is necessary to pay attention to yellow and blue products; because of hue deflection, a light source with a low color temperature causes the hues of yellow and blue products to deflect to red.

For bright spray-painted plastic products, illumination levels only affect the chroma of yellow plastic products. However, CCT greatly affects green and yellow color levels. One possible explanation is the visual effect of the human eye. In 1924 and 1951, the Commission internationale de l'éclairage (CIE) published a curve, regarding the response of the human eyes to different wavelengths. The wavelength to which the human eye is most sensitive is 507 nm (yellowish green), whereas the wavelengths to which the human eye is less sensitive are 400 nm (blue) and 700 nm (red) [46]. The interaction between illuminance and CCT engenders a significant difference in people's perceptions of the lightness, chroma and hue of green products and the hue of red products. The results of this study confirm that the illuminance and correlated color temperature of an LED light source interact with the colors green and yellow. Similar results can also be found in a previous study, conducted by Wu et al., [35], which determined that a person's accuracy in discriminating the color of yellow in LED-lit environments is higher than for magenta and cyan.

For yellow products, a bright spray-painted surface is more affected by the light source than a matte spray-painted surface. One possible explanation is that when a color sample's SRD is close to a spike in a light source's SPD, such similarities create larger perceived color differences [47]. As evident in Figure 5, although the SRDs of the bright spray-painted and matte spray-painted yellow stimulus are close to each other, the SRD of bright paint stimulus is closer to the light source's SPD than that of matte painted one. For the red product, although the SRD of the bright spray-painted sample is closer to the light source's SPD, the human eye is less sensitive to red (at 700 nm), explaining why similar results were not obtained for bright and matte spray-painted stimuli.

### 4.1. Recommended Light Source Settings for the Four Colors

Different LED light source settings can be used at various illuminance levels and CCTs to effectively illuminate ABS plastic products with different surfaces and colors by considering the differences between the lightness, chroma, and hue of the colors. For matte spray-painted green plastic products, the hues of the four LED-lit environments deflect to yellow, but their lightness increased, under the conditions of 4000 K * 1500 lx and 4000 K * 500 lx, and decreased, under the conditions of 2800 K * 500 lx (Figure 6). It is thus recommended to use the light source of 2800 K * 1500 lx when displaying green matte spray-painted products. For almost all yellow products, the four LED-lit environments resulted in lightness reduction and a red-deflected hue. Therefore, a 4000 K * 1500 lx light setting is recommended, due to its comparatively minimal effects on color. The hue of blue products shifted to red, and their lightness decreased, under the conditions of 2800 K * 1500 lx and 2800 K * 500 lx. It is thus recommended to use light source settings that are either 4000 K * 1500 lx or 4000 K * 500 lx. The hue of red products shifted to yellow, and their lightness increased, under all four light source settings; the setting 2800 K * 1500 lx had the greatest effect, thus, we recommend avoiding this setting.

For bright spray-painted green plastic products, color lightness was low under all four LED-lit environments, and their hue was skewed toward yellow. However, 4000 K * 1500 lx and 4000 K * 500 lx conditions yielded the lowest perceived color difference. It is thus suggested to use this light source setting when displaying green colored products. For yellow products, lightness under all four LED-lit environments was low, and hues were all skewed toward red. We recommend using a 4000 K * 1500 lx light source setting, due to its minimal effects on color. For blue products, the lightness of the four LED-lit environments was low, and hues deflected to red. The light source setting of 4000 K * 1500 lx had the least effect on color. For red products, lightness was high, and hue deviation was small under all four LED-lit environments. It is thus recommended to use light sources set at 4000 K * 1500 lx and 2800 K * 1500 lx.

### 4.2. Limitations and Future Suggestions

To ensure that poor eyesight, resulting from old age, did not compromise the validity of the study, subjects in this study were between 20 and 30 years old [48]. Other age groups should be studied in future. Another limitation of this study is that it mainly focused on ABS plastics. Although ABS plastics are widely used in product design, other plastics should be studied in the future, such as polyethylene terephthalate (PET), which is the main material used to make plastic beverage bottles.

### 5. Conclusions

In this study, we adjusted CCT and illumination to create four LED-lit environments. In these environments, 20 participants viewed the color of a standard stimulus and compared the color with a set of stimuli under the standard light source D65.

To ensure the validity of the results, many experimental details were controlled. First, the participants were mainly young people, who passed vision and color blindness tests, which ensured the validity of the experimental results. Second, the light source was adjusted with a photometer and colorimeter to ensure that the settings were within the designated values before the experiment

commenced. Finally, the height of the chair was adjusted individually, to ensure that the visual angle was set within 30°.

MANOVA was used for statistical analysis, and the results indicated two main findings. First, CCT produced greater differences in perceived color than illuminance. Second, for both matte and bright spray-painted ABS plastics, the effects of LED sources on green and yellow ABS plastics were more obvious than their effects on red and blue plastics. This may be related to the sensitivity of the human eye. Additionally, the color difference tendencies of matte and bright spray-painted ABS plastic products are summarized. For matte spray-painted ABS plastic products, under a high CCT, the lightness of green and blue colors increased, but the lightness of yellow decreased. The lightness of red increased under all four LED-lit environments. The hues of yellow and blue shifted to red at low CCTs. Green and red colors shifted toward yellow under all four LED-lit environments. For bright spray-painted ABS plastic products, under the four LED-lit environments in this experiment, the lightness of green, yellow and blue colors decreased, whereas the lightness of red colors increased. Under low CCTs, green hues deflected toward yellow, yellow hues deflected toward red, and blue hues were deflected toward red. The results of this study will allow designers to reduce color differences when matching a product to its target display atmosphere, and—more crucially—help improve the role of LED lighting in sustainable development.

**Author Contributions:** X.D.; writing—original draft preparation, C.-F.W.; supervision, project administration, K.-C.L.; investigation, J.-J.L.; data analysis, investigation.

**Funding:** The study was financial supported by Ministry of Science and Technology, Taiwan (grant number MOST 105-2221-E-036-006-MY2).

**Conflicts of Interest:** The authors declare no conflict of interest.

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
