# Peer review of "What Color Does the Consumer See? Perceived Color Differences in Plastic Products in an LED-Lit Environment"

_sustainability, doi:10.3390/su11215985_

Round 1

Reviewer 1 Report

The paper is well-written and structured and it is also relevant to the 'Sustainability' journals readership.

The abstract is clear and it provides strong interrelations between the title and main research objectives.

I found that the narrative order of this paper has been exquisitely designed by the authors.

The authors were undertaking several multivariate analysis statistical tests which elaborate several outcomes to prove their theoretical approach.

The methodology is novel and it is well grounded to fill a knowledge gap in this specific research field.

Although, therefore, my main concern is that the conclusion does not reflect the quality of data interpretation and discussions have been made by the authors.

I recommend to the authors to improve the conclusion substantially and devise a sub-section and interpret limitations have been experienced throughout setting up the methodology and technical challenges experienced to run the statistical analysis. 

Author Response

Response to Reviewer A Comments

Thank you for your acknowledgment of the study and your suggestions for the conclusions section.

Your suggestions are addressed as follows:

Point 1: I recommend to the authors to improve the conclusion substantially and devise a sub-section and interpret limitations have been experienced throughout setting up the methodology and technical challenges experienced to run the statistical analysis.

Response 1:

After carefully considering your suggestions, we added a new paragraph to the conclusion and adjusted the word sequence and content. You may find the revisions in red in the conclusion. (line 367-378, p14)

According to your suggestions, the new paragraph describes limitations and key details of the methodology, including selecting participants, correcting the lighting source, and adjusting participants’ view angles before the experiment. These details were added because they are crucial to the validity of the results and deserved to be summarized in the conclusion.

In the next paragraph, the results of the study are summarized. First, CCT produced differences in perceived color that were larger than those produced by illuminance. Second, the effects of LED sources on green and yellow ABS plastic were more obvious than those on red and blue. Subsequently, a brief summary of the color differential tendencies was added.

We hope you determine that the conclusion-related revisions meet the requirements for publication.

Reviewer 2 Report

Wu et al submitted their manuscript entitled “What color does the consumer see? Perceived color differences of plastic products in an LED-lit environment” for consideration for publication in Sustainability. The Introduction section is very written that the authors clearly present the background information, previous research results on this area by other research groups and the motivation of this study. The English is also good which makes reading the manuscript enjoyable. The design of the experiments is good and the conclusions are well supported by the amount of data in this work. Therefore, the reviewer suggests the manuscript be accepted in the current form.

One minor query about the experimental setup shown in Figure 1.6a is that although the height of the chair (45 cm) and the table (80 cm) are set to be a constant, due to different sizes of human subjects the distance between the light spot in the box and the human eyes may have little variation. How would the authors respond to this question?

Wu et al submitted their manuscript entitled “What color does the consumer see? Perceived color differences of plastic products in an LED-lit environment” for consideration for publication in Sustainability. The Introduction section is very written that the authors clearly present the background information, previous research results on this area by other research groups and the motivation of this study. The English is also good which makes reading the manuscript enjoyable. The design of the experiments is good and the conclusions are well supported by the amount of data in this work. Therefore, the reviewer suggests the manuscript be accepted in the current form.

One minor query about the experimental setup shown in Figure 1.6a is that although the height of the chair (45 cm) and the table (80 cm) are set to be a constant, due to different sizes of human subjects the distance between the light spot in the box and the human eyes may have little variation. How would the authors respond to this question?

Author Response

Response to Reviewer B Comments

Thank you for your acknowledgment of the study.

Your comments are addressed as follows:

Point 1:One minor query about the experimental setup shown in Figure 1.6a is that although the height of the chair (45 cm) and the table (80 cm) are set to be a constant, due to different sizes of human subjects the distance between the light spot in the box and the human eyes may have little variation. How would the authors respond to this question?

Response 1:

Our apologies for the missed details. We briefly mentioned the adjustment of the participants view angles in the original text. However, the details were not provided in full.

The standard height of the chair was 45 cm. As illustrated in Fig. 1, the chair was adjustable in height, and each participant adjusted the chair height.

Because the distance between the center of the seat and the desktop was fixed (42.3 cm), as long as the height between the participant’s eyes and the desktop was measured, the view angle could be calculated within 30°.

To make the details more complete, we improved Fig. 1a by adding “adjustable height” next to “45 cm.” We also added a revision to the manuscript in red. (line 158-160, p6)

We hope you determine that the revisions meet the requirements for publication.